# The Association between Dietary Iron Intake and Incidence of Dementia in Adults Aged 60 Years or over in the UK Biobank

**DOI:** 10.3390/nu15020260

**Published:** 2023-01-04

**Authors:** Jiahao Liu, Yutong Chen, Xi Lu, Xiaojing Xu, Gabriella Bulloch, Susan Zhu, Zhuoting Zhu, Zongyuan Ge, Wei Wang, Xianwen Shang, Mingguang He

**Affiliations:** 1Faculty of Medicine, Dentistry and Health Sciences, University of Melbourne, Melbourne, VIC 3010, Australia; 2Centre for Eye Research Australia, Royal Victorian Eye and Ear Hospital, Melbourne, VIC 3002, Australia; 3Faculty of Medicine, Nursing and Health Science, Monash University, Clayton, VIC 3800, Australia; 4Charles Perkins Centre, Faculty of Medicine and Health, University of Sydney, Sydney, NSW 2006, Australia; 5Melbourne School of Population and Global Health, University of Melbourne, Melbourne, VIC 3010, Australia; 6Austin Hospital, University of Melbourne, Melbourne, VIC 3084, Australia; 7Monash e-Research Center, Faculty of Engineering, Airdoc Research, Nvidia AI Technology Research Center, Monash University, Melbourne, VIC 3800, Australia; 8State Key Laboratory of Ophthalmology, Zhongshan Ophthalmic Center, Sun Yat-sen University, Guangzhou 510060, China; 9Guangdong Eye Institute, Department of Ophthalmology, Guangdong Provincial People’s Hospital, Guangdong Academy of Medical Sciences, Guangzhou 510080, China

**Keywords:** dietary iron, dementia, gender, modification analysis, sensitivity analysis

## Abstract

**Background** Several studies have investigated the association between dietary iron intake and cognitive impairment, but little is known about the relationship between iron intake and dementia incidence. **Objectives** This study explored the association between dietary iron intake and incident dementia in males and females. Whether this association was modified by factors such as age and medical diseases was also examined. **Methods** We included 41,213 males and 48,892 females aged 60 years or over, from the UK-Biobank cohort. Dietary iron intake was measured using a web-based 24-h dietary recall questionnaire from between 2009 and 2012. Incident dementia was ascertained using hospital inpatient records and death registers until April 2021. Cox proportional regression models examined the association between iron intake and incident dementia, and hazard ratio curves were constructed with knots from the analysis indicating insufficient or excessive iron intake. **Results** During a mean follow-up of 11.8 years, 560 males and 492 females developed dementia. A non-linear relationship between iron intake and incident dementia was observed in both males and females. The lowest incidence rates were observed in the higher iron intake quintile (Q4: ≥15.73, <17.57 mg/day) for males, and the intermediate iron intake quintile (Q3: ≥12.4, <13.71 mg/day) for females. Among those aged 60 and above, all-cause dementia in males was associated with deficient iron intake (Q1 versus Q4: Hazard ratio [HR]: 1.37, 95% Confidence interval [95%CI]: 1.01–1.86, *p* = 0.042) and excessive iron intake (Q5 versus Q4: HR: 1.49, 95%CI: 1.14–1.96, *p* = 0.003), whilst significant associations between all-cause dementia and deficient iron intake were only observed in females without hypertension. Smoking status was a significant moderator (*p*-value for trend = 0.017) for dementia in males only. **Conclusions** Excessive iron intake (≥17.57 mg/day) is associated with a higher incidence of all-cause dementia in males and smoking status modified this association amongst males. Deficient iron intake (<10.93 mg/day) was associated with a higher incidence of all-cause dementia in females without a history of hypertension.

## 1. Introduction

Dementia is a neurodegenerative disease of ageing associated with significant morbidity, mortality, and health care burden. Alzheimer’s disease (AD) and vascular dementia (VD) are subtypes of dementia responsible for 60–70% and 15% of all dementia cases, respectively [1,2]. Approximately 55 million people worldwide suffer from dementia, and this increases by 10 million per year [3]. It is estimated 12.5% of all deaths in England and Wales in 2019 were attributable to dementia [4], and the disease lowers quality of life and requires high levels of care [5]. The total global cost cause of dementia was USD 818 billion in 2015 and will continue to rise in the future, imposing a tremendous burden on healthcare systems worldwide [2]. AD and VD accompany frontal lobe myelin loss, perivascular space dilation and cortical atrophy, which contribute to cognitive decline [6,7]. This cognitive decline may predate diagnosis by 20 years or more, by which time the damage is irreversible [8]. 

As there is currently no effective treatment to stop the progression of dementia, modifiable factors such as dietary intake are essential for the prevention and understanding the pathogenesis of the disease. Some studies show a higher intake of vegetables, fruits and fish is associated with a lower risk of cognitive impairment [9,10], while excessive intake of red meat, processed meat, salt, sugar and saturated fat is positively associated with an increased risk of dementia [11,12,13,14]. Previous studies have demonstrated that iron deficiency is often present in patients with hypertension, diabetes, and stroke, which are well-known risk factors for dementia [15,16,17]. In addition, males have higher brain iron levels compared to females throughout their lifespan, and these differences remain several years after menopause [18]. The recommended dietary iron level is substantially different for males and females, with the average recommended dietary iron intake being 19.3–20.5 mg/day for adult males and 17.0–18.9 mg/day for adult females [19]. These differences are attributed to gender-specific responses to iron metabolism gene variants, transferrin C2 (TfC2) and hemochromatosis (HFE) H63D and HFE C282Y, which are responsible for iron absorption, iron transportation and iron affinity regulation, respectively [18,20,21]. Previous studies demonstrated mutations on these are protective against iron deficiency anemia for reproductive-age females but not in males [21], and HFE C282Y mutations accompanied by haem iron intake is associated with higher ferritin levels in post-menopausal females [22]. Gender-specific responses to iron metabolism gene variants may be a theoretic underpinning for the association between iron intake and dementia in females and males separately.

To date, several studies have linked dietary iron intake to Parkinson’s disease dementia or cognitive function in humans, but none examined all-cause dementia, AD or VD [23,24,25,26]. Unfortunately, many studies are subject to recall bias and reversal causational bias, and the non-linear relationships between iron intake and cognitive function were overlooked. Therefore, this study examined linear and non-linear relationships between dietary iron intake and the incidence of all-cause dementia. AD and VD were investigated in UK Biobank participants aged 60 years or over. We also examined the effects of gender, age, education, smoking and other medical conditions.

## 2. Method 

This study utilised data from the UK Biobank population-based cohort. More than half a million people aged 40–69 years participants from the United Kingdom were recruited between 2006 and 2010 [27]. Participants answered touch-screen self-reported questionnaires, computer-assisted personal interviews online or attended the allocated 22 assessment centres to complete the physical measurements in England, Scotland and Wales [27]. Analyses were restricted to individuals aged 60 years or over (as most incident dementia cases occur in older adults). To prevent reverse causality bias, participants who were diagnosed with dementia before the diet survey examination were excluded.

### 2.1. Ethics

The UK Biobank Study’s ethical approval had been permitted by the National Information Governance Board for Health and Social Care and the NHS North West Multicentre Research Ethics Committee. Participants’ consent is required to be signed at recruitment. This study was established under application number 62,443 of the UK Biobank resource.

### 2.2. Sample Selection Flowchart

A total of 502,505 subjects were examined at baseline. We excluded subjects who had an incomplete iron intake record or iron intake ≤0 mg/day, were <60 years of age, missing metabolic rate data, under- or over-reporting energy intake (under-reporting: <1.1 × metabolic rate—500 kcal; over-reporting: 2.5 × metabolic rates + 500 kcal); energy-adjusted iron intake ≤0 mg/day; stroke and dementia at baseline; or a missing follow-up dementia record. After exclusion criteria were applied, a remaining 41,213 male and 48,892 female subjects were eligible for analysis.

## 3. Measurement Method

### 3.1. Dementia Data

Dementia (all-cause dementia, AD, VD) was defined using hospital inpatient records at the Hospital Episode Statistics for England, Scottish Morbidity Record data for Scotland, and the Patient Episode Database for Wales. The diagnostic tools used were the *International Classification of Disease version 10 (ICD-10)* coding system, and the primary/secondary care data were diagnosed by the *Read coding system (version 2 or 3)*, which is operated by qualified and trained professional recorders. Person years were recorded based on the onset of dementia, date of death, or the end of follow-up (28 April 2021) [27]. Baseline was defined as the date of the last dietary assessment.

### 3.2. Dietary Intake

Dietary intake was assessed using an online self-administered questionnaire system within 24 h (based on the EPIC-soft 24-h recall questionnaire). The dietary survey assessments were completed through an online questionnaire on one or more of the five occasions between April 2009 and June 2012 [27]. Potential recall bias was minimised by timely data collection. The percentages of participants with one to five dietary surveys were 36%, 23%, 22%, 16% and 3%, respectively, for males. The corresponding values for females were 37%, 23%, 21%, 15% and 3%, respectively. 

Intakes of foods, including red meat (servings/day), processed meat (servings/week), fish (servings/week), vegetable (servings/day), fruit (servings/day) and iron complementary (Yes/No) were computed by averaging the number of assessment surveys. Nutrients, including iron (mg/day), energy (kcal/day), sugar (g/day), vitamin C (mg/day) vitamin E (mg/day), fat (g/day), saturated fat (g/day) and polyunsaturated fat (g/day) were computed based on built-in algorithms and the food composition data of the *Composition of Foods 6th edition (2002)* [28]. Iron intake for model analysis were extracted from dietary foods. Iron supplements were not included as part of iron intake. The average iron was calculated for those with two or more assessments. Furthermore, gender-specific energy-adjusted iron intake was computed using the formula of residuals for the subject from a regression model, with iron intake and total energy intake as the independent variable plus the expected iron intake for a person with mean energy intake [29]. Afterwards, energy-adjusted iron intake was categorised by quintiles and the linear and restricted cubic spline construction.

### 3.3. Covariates

Demographic information was self-reported, including gender (female, male); age (years); ethnicity (Whites, non-whites); income (<GBP 18,000, GBP 18,000–30,999, GBP 31,000–51,999, GBP 52,000–100,000, >GBP 100,000, do not know, prefer not to answer); and education (high level [college or university degree (Level 6–8 education qualifications)], intermediate level [A levels/AS levels or equivalent, O levels/GCSEs or equivalent, CSEs or equivalent, NVQ or HND or HNC or equivalent, and other professional qualifications (Level 2–5 education qualifications)], and low level [none of the aforementioned]). Lifestyle factors included physical activity (Metabolic equivalents minutes/week [MET-min/week]); sleep duration (≥7 h/day and ≤9 h/day or ≤7 h/day and ≥9 h/day); smoking status (never, former, current, prefer not to answer); alcohol consumption (never, former, current, prefer not to answer); and were collected using a touch-screen self-report questionnaire [27]. Medical history of cardiovascular disease (CVD), hypertension, depression, and diabetes, were defined using a self-reported questionnaire and nurse’s interview [27] and determined as covariates. Metabolic rate (kcal/day) was calculated by the Oxford equation [30], and weight and height were measured by a Tanita BC-418 MA body composition analyser [27]. Body mass index (BMI) (kg/m^2^) was computed by weight in kilograms divided by height in meters squared. Glycated haemoglobin (HbA1c) (mmol/mol) was measured by the VARIANT II TURBO Hemoglobin Testing System and Bio-Rad. Blood cholesterol was measured by AU5400 and Beckman Coulter [31]. Apolipoprotein Ɛ4 (APOE4) genotype was defined by Affymetrix using a BiLEVE Axiom array [32]. Further information is available on the UK-Biobank Web site (https://www.ukbiobank.ac.uk (accessed on 28 April 2021)) [33]. In this research, there are no imputed values for the missing covariates.

## 4. Statistical Analysis

Data of baseline characteristics were presented as frequency (percentage) and mean (standard deviation) for gender. X^2^ tests for categorical covariates and variance tests (ANOVA) for continuous covariates were used to test the difference in baseline characteristics across the quintiles of energy-adjusted iron intake.

Cox proportional risk regression models estimated continuous hazard ratios (HR) for dementia associated with energy-adjusted iron intake. This analysis was conducted in males and females separately.

Iron intake was categorised into quintiles (Q1: deficient intake, Q2: lower intake, Q3: intermediate intake, Q4: higher intake, Q5: excessive intake). The iron intake quintile with the lowest incidence rate was assigned as the reference group for Cox proportional hazard regression models. Additionally, iron intake quintiles were applied to stratification analysis and moderation analysis. We analysed all-caused dementia, AD and VD subtypes for males and females separately, as they differed in iron intake requirement.

Model 1 (M1) was adjusted for age (continuous). Model 2 (M2) was adjusted for M1 plus BMI (continuous), smoking (categorical), alcohol intake (categorical), income (categorical), education (categorical), physical activity (continuous), hypertension (categorical), CVD (categorical), diabetes (categorical), blood cholesterol (continuous), depression (categorical), energy intake (continuous), ethnicity (categorical), APOE4 status (categorical) and sleep duration (categorical). Model 3 (M3) was adjusted for M2 plus intake of salt (categorical), red meat (continuous), processed meat (continuous), fish (continuous), vegetable (continuous), fruit (continuous), sugar (continuous), vitamin C (continuous), vitamin E (continuous), fat (continuous), saturated fat (continuous) and polyunsaturated fat (continuous) and iron supplement (categorical).

We examined whether the association between iron intake and all-cause dementia was moderated by covariates such as demographic (continuous age, categorical age: <65 years or ≥65 years), smoking status (pseudo-continuous; categorical: current smoker, previous smoker, never smoke) and medical conditions (diabetes (Yes/No), hypertension (Yes/No), CVD (Yes/No), depression (Yes/No)) and APOE4 (Yes/No)). 

Moreover, we examined the association between dietary iron and all-cause dementia stratified by covariates included in the moderation analysis. Furthermore, we conducted a sensitivity analysis to examine the association between iron intake and incident dementia in participants with two-or-more and three-or-more dietary assessments.

The data preparation software we used in this study was STATA version 16.0 (StataCorp LLC, College Station, TX, USA), and statistical analysis was operated by R version 4.0.3 (The R Foundation for Statistical Computing c/o Institute for Statistics and Mathematics, Vienna, Austria) coded by RStudio Desktop 1.4.1106 (Boston, MA, USA).

## 5. Results

### 5.1. Baseline Characteristics 

Table 1 shows baseline characteristics according to the quintile of energy-adjusted iron intake (Q1–5). 

Participants in higher quintiles had higher physical activity levels (MET-min/week), and higher intakes of red meat, fish, vegetable, fruit, and Vitamin D supplements. Meanwhile, participants in lower quintiles had a higher intake of processed meat intake, salt and saturated fat, higher BMI (kg/m^2^), and most slept between 7 and 9 h. Individuals in Q1 and Q5 of iron intakes had a higher energy intake (kcal/day) compared to the intermediate quintiles. In addition, participants in Q1 had the highest proportion of those with intermediate education level, and income <18,000 GBP/annual, were current smokers, were never or previous alcohol drinkers, and had diagnosed depression (Table 1).

### 5.2. Population Selection

Of 502,505 participants, a total of 41,213 male (45.7%) and 48,892 female (54.3%), aged 60 or over (mean ± SD: male: 65.6 ± 3.4; female: 65.2 ± 3.3) subjects were included in the analysis (Figure 1 and Table 1).

### 5.3. Incidence of Dementia

There were 479,010 person-years and 574,303 person-years for males and females, respectively (median follow-up period for males: 11.8 years; females: 11.8 years). Amongst 41,213 males, 560 were diagnosed with all-cause dementia (247 incident AD cases and 117 incident VD cases) at follow-up. Of 48,892 females, 492 were diagnosed with all-cause dementia (228 incident AD cases and 77 incident VD cases) (Table 2). The incidence of all-cause dementia was higher in males (Incidence rate [IR]: 11.7 per 10^4^ person-year [py]) than in females (IR: 8.6 per 10^4^ py).

### 5.4. Iron Intake and Incident Dementia

U-shape associations between the incidence of dementia and iron intake were observed in both genders and increased sharply as iron intake decreased, particularly In females (Figure 2). Q4 and Q3 had the lowest incidence of dementia and were selected as the reference groups for males and females, respectively.

The HR was 1.37 (95%CI: 1.01, 1.86, *p* = 0.042) for Q1 versus Q4 and 1.49 (95%CI: 1.14, 1.96, *p* = 0.003) for Q5 versus Q4 in the multivariable-adjusted analysis in males. In contrast, iron intake and the incidence of all-cause dementia showed no statistical significance in females (Table 2).

### 5.5. Moderation Analysis

Moderation analyses were conducted to examine whether the association between iron intake and all-cause dementia differed across subgroups of genders: age, smoking status, diabetes, hypertension, CVD, depression and APOE4. Smoking status was a significant moderator in males (M2: *p*-value for trend = 0.018). Hypertension status only modified the association between all-caused dementia and deficient iron intake among females.

### 5.6. Stratification Analysis

In males, a significant U-shape association was found in those who had no CVD history (M3: Q1:Q4: 1.53 (1.1, 2.12), *p* = 0.012; Q5:Q4: 1.51 (1.12, 2.04), *p* = 0.007) and no depression (M3: Q1:Q4: 1.48 (1.08, 2.02), *p* = 0.014; Q5:Q4: 1.48 (1.11, 1.97), *p* = 0.007). Excessive iron intake was associated with an increased risk of all-cause dementia in those aged ≥65 years old (M3: HR (95% CI): Q5:Q4: 1.92 (1.10–3.35), *p* = 0.021), without hypertension (M3: Q5:Q4: 1.44 (1.01, 2.05), *p* = 0.041), without diabetes (M3: Q5:Q4: 1.46 (1.09, 1.95), *p* = 0.011) and non-APOE4 carriers (M3: Q5:Q4: 1.53 (1.05, 2.23), *p* = 0.027). In contrast, deficient iron intake was associated with an increased risk of all-cause dementia in those with smoking history (M3: Q1:Q4: 1.53 (1.02, 2.30), *p* = 0.041) and hypertension (M3: Q1:Q4: 1.70 (1.07, 2.72), *p* = 0.025).

For females, lower iron intake was associated with a higher risk of all-cause dementia in individuals without hypertension (M3: Q1:Q4: 1.47 (1.01, 2.15), *p* = 0.044).

### 5.7. Sensitivity Analysis

Among males who had ≥2 dietary assessments (n = 26,505, number of dementia events = 279), excessive iron intake was associated with an increased incidence of all-cause dementia (M3: Q5:Q4: 1.67 (1.14, 2.46), *p* = 0.009). Conversely, no significant association was observed amongst females who completed ≥2 dietary assessments (n = 30,585, number of dementia events = 261).

## 6. Discussion

In this longitudinal study, a U-shaped relationship between iron intake and incident dementia was seen across males and females, indicating that both deficient and excessive iron intake are associated with an increased risk of dementia. For males ≥60 years, excessive iron intake (Q5: ≥17.57 mg/day) was associated with incident dementia compared with the reference iron intake (Q4: ≥15.73, <17.57 mg/day) population, whilst no significant association was observed between dietary iron intake and incident dementia in females. Additionally, we observed smoking status increased the association between dietary iron intake and incident dementia in males. Lastly, the incident dementia was associated more with deficient iron intake in females with hypertension.

Our data suggest males with excessive iron intake may increase the incidence of dementia if they have one of the following features: (1) aged >65 years, (2) non-hypertension, (3) non-diabetes, and (4) non-APOE4 genetic factor. In contrast, deficiency and decreased iron consumption may increase the incidence of dementia for those with hypertension. A U-shape association had been found in males with one of the following features: (1) aged ≥60 and <65 years, (2) ever smoker, (3) without CVD or, (4) depression.

In females, hypertension negatively modified the association between deficient iron intake and the incidence of dementia. Additionally, with no hypertension, deficient iron intake significantly increased the incidence of dementia compared with intermediate iron intake.

Iron homeostasis is tightly regulated by the hormone hepcidin, which induces the translation of iron channels and regulated the blood–brain barrier (BBB) [7,34]. A mechanical cause for iron’s role in neurodegeneration stems from bypassing these two gatekeepers through an initiating cause that disrupts the BBB, such as endothelial dysfunction. Subsequently, iron seeps into the brain parenchyma, allowing for excessive iron deposition as it bypasses iron transporters on the BBB regulated by hepcidin. Previous vivo and vitro studies demonstrate that iron accumulation and ferroptosis strongly correlate with neuroinflammation, myelination, and neurodegenerative protein synthesis [34,35,36,37,38,39]. Potentiation of these pathways is associated with mitochondrial energy transduction, enzyme catalysis, mitochondrial function, α-synuclein synthesis, β-amyloid (Aβ) misfolding, and plaque aggregation [35,36,37,38,39]. Furthermore, unstable forms of hemosiderin and oxyhydroxides resulting from iron dyshomeostasis may induce oxidative stress and subsequent tau phosphorylation, which has also been recognised as a critical player in the pathogenesis of neurodegeneration [10,34,36,40,41].

In males, both deficient and excessive iron intake increased the incidence of all-cause dementia for males aged ≥60 years and above, and the association remained robust following sensitivity analysis. In the stratification analysis, intermediate to deficient iron intake increased the incidence of early-onset dementia (<65 years), whilst excessive iron intake elevated the risk of dementia among adults aged ≥65 years. Previous research explains that male-exclusive allelic variants for iron metabolism proteins, hemochromatosis H63D (HFE H63D) and plasma transferrin C2 (TfC2) play vital roles in neuronal iron delivery [18,34], and Naets et al. highlighted that testosterone could enhance iron absorption and erythropoiesis [42]. In addition, age-related iron shifts are associated with iron accumulation in the brain [43]. A previous cohort study found that increased iron intake was associated with cognitive decline in the Chinese population [24], and remained robust across broader age ranges. These findings imply iron intake should reduce with age to minimize modifiable risk, although it should be noted those with stroke history were not excluded from the current analysis. Conversely, case-control studies conducted in Japan and the United States found an inverse relationship between iron intake and the risk of cognitive decline and Parkinson’s disease [26,44,45]. These seemingly contradictory results could be explained by the U-shaped association discovered between incident dementia and dietary iron intake in the current study.

This study found that smoking modified the association between dietary iron intake and the incidence of dementia in males. In males with a history of smoking, both deficient and excessive iron intake was associated with incident dementia. Smoking is known to influence iron transporters, and nicotine suppresses iron absorption via impeding iron transmission from transferrin and endocytosis [46]. Previous studies by Durazzo et al. and Pirpamer et al. established that former smoking significantly increased the risk for AD from the damage caused by smoking-related cerebral oxidative stress, and identified this as a significant risk factor for cerebral iron accumulation, a habit that is also known to initiate endothelial dysfunction [47,48]. Another cross-sectional study by Li et al. found significantly higher cerebrospinal fluid (CSF) iron concentration in active smokers than in non-smokers, suggesting that smoking may accelerate cognitive impairment [49]. The present study further propagates this knowledge by providing insight into how smoking alters the relationship between dietary iron intake and the incidence of dementia.

Both deficient and excessive iron intake increased dementia incidence among males without CVD or depression in the current study, while excessive iron intake increased dementia incidence among males without diabetes or non APOE4 carriers. These results are counterintuitive to preceding studies which revealed that low serum transferrin receptor and serum ferritin binding ratio significantly increased the risk of CVD in males [50], and lower transferrin saturation was negatively associated with dementia [45]. In addition, meta-analyses have previously identified associations between depression and dementia, and diabetes with AD [51,52]. Moreover, cortical iron aggregation and APOE4 act synergistically with cognitive network activity, including brain regions such as the medial prefrontal cortex, lateral parietal cortex, posterior cingulate and hippocampus [53]. The results of this study imply that dietary iron intake should be considered a risk factor for dementia in males independently of these medical and physiological risk factors; however, the confounding results of studies imply that further research should be conducted before this association is considered generally.

This study highlights the many gender differences between dietary iron intake and the incidence of dementia. We did not observe a significant association between dietary iron intake and incident dementia in the female population, which is consistent with a longitudinal study from the United Kingdom that proposed iron intake and incident dementia had sex-specific differences [54] although in this study participants with stroke history were also not excluded. There is evidence demonstrating that more than half of patients after a stroke had acquired post-stroke neurocognitive disorder [55]. In contrast, Kezele et al. observed that advanced dementia was only associated with lower iron levels in adolescent females [34], and both Power et al. and Cherbuin et al. observed increased dietary iron intake was associated with risk for Parkinson’s disease and cognitive impairment [25,26]. An explanation for this association is that females often have a higher vegetable intake and lower red meat intake compared to males, which is associated with greater central grey matter in the thalamus, pulvinar nucleus and red nucleus, and a lower level of iron accumulation. Females in menarche (19.6 mg/day) have a significantly higher daily intake than males (9.1 mg/day) [20], and menopause is significantly associated with a two-to-threefold rise in serum iron and iron storage despite serum concentration still being considered within the physiologic range [25,56]. Losses in oestrogen, which curtail improper storage of iron by downregulating hepcidin, may potentiate storage and induce neurodegenerative changes in older age [57]. Although these uncover some insight to sex-specific differences and possible mechanisms, common limitations of these studies were recall bias and reverse causality bias in the study design, leaving more research necessary to validate these claims.

This study revealed that hypertension negatively mediated the association between deficient iron intake and dementia incidence in females, while hypertension exaggerated this association in males. Meanwhile, non-hypertensive males with excessive iron intake were more likely to acquire dementia. The result is inconsistent with Shi et al., who demonstrated that the presence of hypertension moderated the association between dietary iron with dementia and cognitive impairment [24], although their analysis did not separate analysis for gender. Despite this, other evidence shows blood pressure is an associated risk of dementia [58], and iron deficiency is more common amongst people with idiopathic pulmonary hypertension compared with disease-free patients [15]. Therefore, a plausible hypothesis is that insufficient iron intake causes hypertension as a byproduct of ensuring sufficient oxygen transport, which then confers to increased risk for dementia. On the other hand, excessive dietary iron is sufficient as a stimulus to increase the risk of dementia independent of hypertension status in males. Thus, the modifying effect of hypertension on the association between dietary iron intake and the incidence of dementia is sex-specific, and future studies should seek to understand the underlying mechanism.

Strengths of this longitudinal study include its large sample size and sensitivity analysis, which minimized bias from missing data and attrition biases. Moreover, dementia cases were identified in agreement with primary care records, and a fundamental validation study was conducted to assess the dementia outcome in UK-Biobank, making dementia as an outcome reliable [59].

Limitations:More than 97.5% of data were obtained from a White Caucasian background leading to poor generalizability to other ethnicities.Geographic differences may also exist between the United Kingdom and other nations, considering the varied prevalence of illness complications and levels of primary health care programmes.Limited age range for subject selection in our investigation, this study has limited generalizability for those under the age of 60.This study used a self-reported questionnaire to calculate iron intake, and this likely leads to measurement errors in iron consumption.Subjects who developed dementia may have delayed reporting or diagnosis, and misclassification bias may have reduced reporting. Likewise, the misclassification of dementia subtypes needs to be considered, as AD and VD often have mixed pathology and pathophysiology [7].Potentially, medications may interfere with iron absorption and impact dementia pathogenesis.There may be insufficient power to establish the synergistic effects or modification effects of the aforementioned medical and physiological conditions. Further studies are needed before a definitive conclusion can be drawn.

## 7. Conclusions

Excessive iron intake (≥17.57 mg/day) is associated with a higher incidence of all-cause dementia in males and history of smoking further increased this risk. Deficient iron intake (<10.93 mg/day) was associated with a higher incidence of all-cause dementia in females without a history of hypertension.

## Figures and Tables

**Figure 1 nutrients-15-00260-f001:**
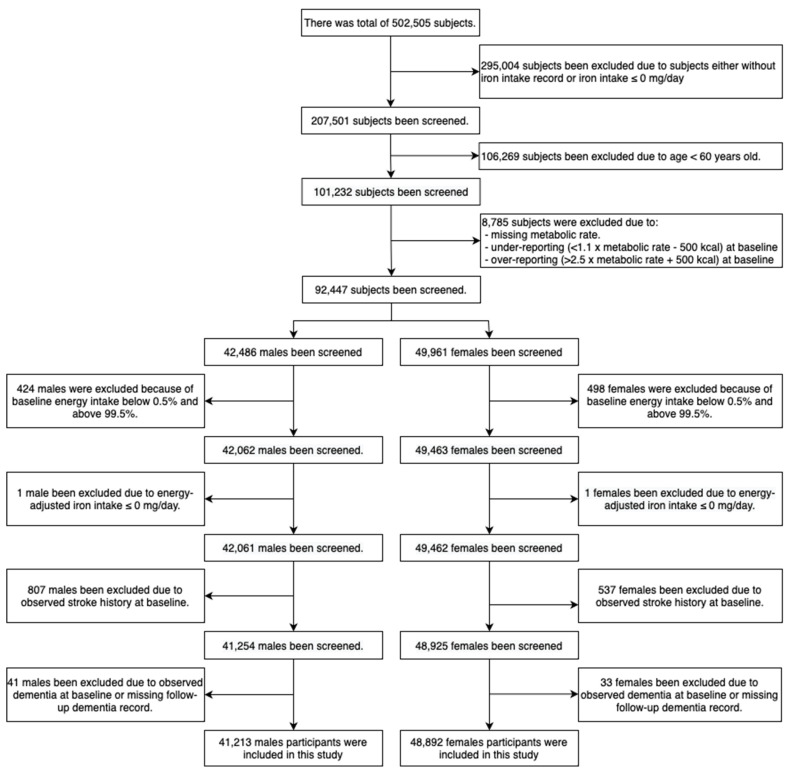
Sample selection flowchart.

**Figure 2 nutrients-15-00260-f002:**
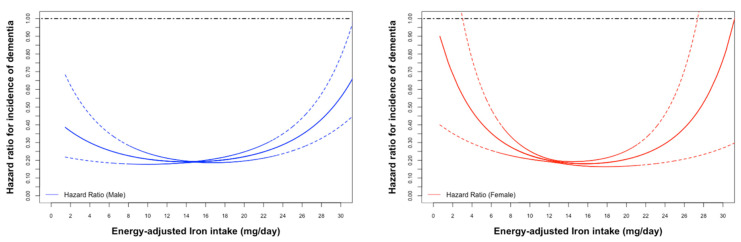
Age-adjusted Cox proportional hazard ratios (HR) (male: blue; female: red) for incidence of all-caused dementia associated with energy-adjusted iron intake.

**Table 1 nutrients-15-00260-t001:** Baseline characteristics according to iron intake. Abbreviations: Q, quintile; n, sample size; SD, standard deviation. The baseline characteristic is presented for different genders (Male/Female). *p*-value is a comparison based on the *X*^2^ test and ANOVA analysis. *X*^2^ tests were conducted for depression, salt intake, iron supplement use, diabetes, cardiovascular disease, hypertension, smoking status, income status, alcohol intake, education, and ethnic background. ANOVA analysis was conducted for age, BMI, blood cholesterol, metabolic rate (kcal/day), dietary intake (total energy, red meat, processed meat, fish, vegetable, fruit, vitamin C, vitamin E, fat, saturated fat, polyunsaturated fat), sleep duration, HbA1c, physical activity.

Male							
	Energy-adjusted iron intake quintiles						
	Total	Q1	Q2	Q3	Q4	Q5	*p*-value
**Range [mg/day]**	1.46, 48.76	<12.59	≥12.59, <14.24	≥14.24, <15.73	≥15.73, <17.57	≥17.57	
**n**	41,213	8243	8243	8242	8243	8242	
**Age (mean [SD])**	65.65 (3.35)	65.39 (3.35)	65.62 (3.37)	65.72 (3.35)	65.76 (3.32)	65.75 (3.34)	<0.001
**Ethnicity (%)**							<0.001
White	40,218 (97.59%)	8007 (97.14%)	8006 (97.12%)	8055 (97.73%)	8090 (98.14%)	8060 (97.79%)	
Mixed	50 (0.12%)	12 (0.15%)	14 (0.17%)	9 (0.11%)	10 (0.12%)	5 (0.06%)	
Non-white	722 (1.75%)	182 (2.21%)	167 (2.03%)	132 (1.60%)	102 (1.24%)	139 (1.69%)	
Do not know/Prefer not to answer	223 (0.54%)	42 (0.51%)	56 (0.68%)	46 (0.56%)	41 (0.50%)	38 (0.46%)	
**Education level (%)**							<0.001
College or University degree	17,342 (42.08%)	2620 (31.78%)	3387 (41.09%)	3670 (44.53%)	3807 (46.18%)	3858 (46.81%)	
A levels/AS levels or equivalent	4435 (10.76%)	843 (10.23%)	894 (10.85%)	895 (10.86%)	912 (11.06%)	891 (10.81%)	
O levels/GCSEs or equivalent	7384 (17.92%)	1694 (20.55%)	1515 (18.38%)	1455 (17.65%)	1385 (16.80%)	1335 (16.20%)	
CSEs or equivalent	659 (1.60%)	164 (1.99%)	127 (1.54%)	142 (1.72%)	115 (1.40%)	111 (1.35%)	
NVQ or HND or HNC or equivalent	3832 (9.30%)	854 (10.36%)	790 (9.58%)	746 (9.05%)	716 (8.69%)	726 (8.81%)	
Other professional qualifications, e.g., nursing, teaching	2319 (5.63%)	486 (5.90%)	473 (5.74%)	446 (5.41%)	464 (5.63%)	450 (5.46%)	
Do not know/Prefer not to answer	5242 (12.72%)	1582 (19.19%)	1057 (12.82%)	888 (10.77%)	844 (10.24%)	871 (10.57%)	
**Income (GBP/annually) (%)**							<0.001
<18,000	6581 (15.97%)	1715 (20.81%)	1342 (16.28%)	1216 (14.75%)	1150 (13.95%)	1158 (14.05%)	
18,000 to 30,999	11,451 (27.78%)	2348 (28.48%)	2365 (28.69%)	2307 (27.99%)	2213 (26.85%)	2218 (26.91%)	
31,000 to 51,999	10,817 (26.25%)	2027 (24.59%)	2192 (26.59%)	2182 (26.47%)	2255 (27.36%)	2161 (26.22%)	
52,000 to 100,000	7088 (17.20%)	1156 (14.02%)	1353 (16.41%)	1447 (17.56%)	1563 (18.96%)	1569 (19.04%)	
>100,000	1787 (4.34%)	217 (2.63%)	298 (3.62%)	399 (4.84%)	417 (5.06%)	456 (5.53%)	
Do not know/Prefer not to answer	3489 (8.47%)	780 (9.46%)	693 (8.41%)	691 (8.38%)	645 (7.82%)	680 (8.25%)	
**Smoking status (%)**							<0.001
Never smoked	18,877 (45.80%)	3492 (42.36%)	3741 (45.38%)	3894 (47.25%)	3891 (47.20%)	3859 (46.82%)	
Ever smoked	22,228 (53.93%)	4725 (57.32%)	4477 (54.31%)	4327 (52.50%)	4333 (52.57%)	4366 (52.97%)	
Do not know/Prefer not to answer	108 (0.26%)	26 (0.32%)	25 (0.30%)	21 (0.25%)	19 (0.23%)	17 (0.21%)	
**Alcohol status (%)**							<0.001
Never drink	742 (1.80%)	210 (2.55%)	158 (1.92%)	135 (1.64%)	122 (1.48%)	117 (1.42%)	
Previous drinker	1064 (2.58%)	320 (3.88%)	234 (2.84%)	174 (2.11%)	159 (1.93%)	177 (2.15%)	
Current drinker	39,381 (95.55%)	7706 (93.49%)	7844 (95.16%)	7930 (96.21%)	7959 (96.55%)	7942 (96.36%)	
Do not know/Prefer not to answer	26 (0.06%)	7 (0.08%)	7 (0.08%)	3 (0.04%)	3 (0.04%)	6 (0.07%)	
**BMI (kg/m^2^) (mean [SD])**	27.15 (3.67)	27.56 (3.86)	27.20 (3.61)	27.06 (3.64)	26.94 (3.55)	26.97 (3.64)	<0.001
**Energy intake (kcal/day) (mean [SD])**	2327 (520)	2410 (581)	2267 (499)	2278 (486)	2287 (481)	2394 (529)	<0.001
**Physical activity (MET-min/week) (mean (SD))**	2613 (2382)	2597 (2490)	2609 (2413)	2582 (2293)	2593 (2307)	2686 (2400)	0.039
**Glycated haemoglobin (HbA1c) (mmol/mol) (mean [SD])**	36.50 (6.42)	36.84 (6.47)	36.46 (6.27)	36.35 (6.18)	36.32 (6.40)	36.51 (6.77)	<0.001
**Blood cholesterol (mmol/L) (mean [SD])**	5.46 (1.09)	5.46 (1.11)	5.47 (1.09)	5.46 (1.08)	5.46 (1.07)	5.43 (1.08)	0.12
**Apolipoprotein E4 (APOE4) (%)**	9342 (23.01%)	1782 (21.93%)	1837 (22.60%)	1852 (22.88%)	1882 (23.16%)	1989 (24.50%)	0.002
**Cardiovascular disease (%)**	3200 (7.76%)	665 (8.07%)	599 (7.27%)	645 (7.83%)	617 (7.49%)	674 (8.18%)	0.15
**Hypertension (%)**	14,222 (34.51%)	2910 (35.30%)	2856 (34.65%)	2804 (34.02%)	2809 (34.08%)	2843 (34.49%)	0.42
**Depression (%)**	1309 (3.18%)	325 (3.94%)	248 (3.01%)	231 (2.80%)	257 (3.12%)	248 (3.01%)	<0.001
**Diabetes mellitus (%)**	2266 (5.50%)	448 (5.43%)	441 (5.35%)	424 (5.14%)	442 (5.36%)	511 (6.20%)	0.033
**Sleep duration = sleep duration ≥7 and ≤9 h (%)**	32,362 (78.52%)	6250 (75.82%)	6452 (78.27%)	6601 (80.09%)	6591 (79.96%)	6468 (78.48%)	<0.001
**Red meat intake (serve/day)** **(mean [SD])**	0.37 (0.45)	0.26 (0.40)	0.33 (0.40)	0.37 (0.42)	0.41 (0.45)	0.47 (0.54)	<0.001
**Processed meat intake (serve/day) (mean [SD])**	0.61 (0.90)	0.73 (1.06)	0.65 (0.90)	0.59 (0.84)	0.57 (0.83)	0.50 (0.83)	<0.001
**Fish intake (serve/day) (mean [SD])**	3.08 (2.31)	2.23 (1.85)	2.68 (1.89)	3.01 (2.00)	3.37 (2.26)	4.10 (2.91)	<0.001
**Vegetable intake (serve/day)** **(mean [SD])**	2.20 (3.95)	1.85 (3.99)	1.99 (3.47)	2.17 (3.70)	2.29 (3.75)	2.70 (4.67)	<0.001
**Fruit intake (serve/day)** **(mean [SD])**	0.33 (0.48)	0.30 (0.48)	0.31 (0.44)	0.33 (0.46)	0.34 (0.47)	0.35 (0.54)	<0.001
**Sugar intake (g/day) (mean [SD])**	129.77 (46.89)	137.63 (54.33)	126.13 (44.34)	126.56 (42.85)	126.52 (43.02)	132.01 (47.84)	<0.001
**Vitamin C intake (mg/day) (mean [SD])**	152.31 (99.34)	118.64 (90.37)	137.75 (87.77)	150.64 (91.68)	164.83 (95.58)	189.69 (114.16)	<0.001
**Vitamin E intake (mg/day) (mean [SD])**	9.47 (4.10)	9.35 (4.48)	8.93 (3.78)	9.27 (3.79)	9.44 (3.87)	10.34 (4.39)	<0.001
**Salt intake (%)**							<0.001
Never/rarely	23,589 (57.24%)	3971 (48.17%)	4597 (55.77%)	4817 (58.44%)	5026 (60.97%)	5178 (62.82%)	
Sometimes	11,005 (26.70%)	2406 (29.19%)	2254 (27.34%)	2214 (26.86%)	2118 (25.69%)	2013 (24.42%)	
Usually	5229 (12.69%)	1364 (16.55%)	1132 (13.73%)	983 (11.93%)	895 (10.86%)	855 (10.37%)	
Always	1377 (3.34%)	498 (6.04%)	259 (3.14%)	225 (2.73%)	203 (2.46%)	192 (2.33%)	
Missing data	13 (0.03%)	4 (0.05%)	1 (0.01%)	3 (0.04%)	1 (0.01%)	4 (0.05%)	
**Fat intake (g/day) (mean [SD])**	84.98 (27.74)	95.00 (29.70)	85.24 (26.16)	83.02 (25.97)	80.46 (26.12)	81.18 (28.02)	<0.001
**Saturated fat intake (g/day) (mean [SD])**	32.93 (12.39)	38.82 (13.87)	33.52 (11.58)	31.93 (11.35)	30.33 (11.21)	30.07 (11.67)	<0.001
**Polyunsaturated fat intake (g/day) (mean [SD])**	15.52 (7.01)	16.76 (7.54)	15.39 (6.75)	15.25 (6.64)	14.96 (6.69)	15.25 (7.25)	<0.001
**Iron supplement (%)**							0.021
No	41,113 (99.76%)	8220 (99.72%)	8232 (99.87%)	8220 (99.73%)	8229 (99.83%)	8212 (99.64%)	
Yes	100 (0.24%)	23 (0.28%)	11 (0.13%)	22 (0.27%)	14 (0.17%)	30 (0.36%)	
**Female**							
	Energy-adjusted iron intake quintiles						
	Total	Q1	Q2	Q3	Q4	Q5	*p*-value
**Range [mg/day]**	0.67, 37.09	<10.93	≥10.93, <12.4	≥12.4, <13.71	≥13.71, <15.39	≥15.39	
**n**	48,892	9779	9778	9779	9778	9778	
**Age (mean [SD])**	65.17 (3.25)	65.05 (3.24)	65.23 (3.28)	65.18 (3.25)	65.25 (3.25)	65.16 (3.22)	<0.001
**Ethnicity (%)**							<0.001
White	47,771 (97.71%)	9483 (96.97%)	9532 (97.48%)	9588 (98.05%)	9584 (98.02%)	9584 (98.02%)	
Mixed	95 (0.19%)	21 (0.21%)	23 (0.24%)	14 (0.14%)	20 (0.20%)	17 (0.17%)	
Non-white	899 (1.84%)	250 (2.56%)	197 (2.01%)	146 (1.49%)	149 (1.52%)	157 (1.61%)	
Do not know/Prefer not to answer	127 (0.26%)	25 (0.26%)	26 (0.27%)	31 (0.32%)	25 (0.26%)	20 (0.20%)	
**Education level (%)**							<0.001
College or University degree	17,420 (35.63%)	2841 (29.05%)	3366 (34.42%)	3621 (37.03%)	3760 (38.45%)	3832 (39.19%)	
A levels/AS levels or equivalent	5944 (12.16%)	1115 (11.40%)	1196 (12.23%)	1249 (12.77%)	1210 (12.37%)	1174 (12.01%)	
O levels/GCSEs or equivalent	12,095 (24.74%)	2588 (26.46%)	2480 (25.36%)	2324 (23.77%)	2414 (24.69%)	2289 (23.41%)	
CSEs or equivalent	1166 (2.38%)	280 (2.86%)	241 (2.46%)	228 (2.33%)	200 (2.05%)	217 (2.22%)	
NVQ or HND or HNC or equivalent	1558 (3.19%)	399 (4.08%)	299 (3.06%)	300 (3.07%)	274 (2.80%)	286 (2.92%)	
Other professional qualifications, e.g., nursing, teaching	3726 (7.62%)	709 (7.25%)	735 (7.52%)	766 (7.83%)	758 (7.75%)	758 (7.75%)	
Do not know/Prefer not to answer	6983 (14.28%)	1847 (18.89%)	1461 (14.94%)	1291 (13.20%)	1162 (11.88%)	1222 (12.50%)	
**Income (GBP/annually) (%),**							<0.001
<18,000	10,153 (20.77%)	2334 (23.87%)	2086 (21.33%)	1914 (19.57%)	1902 (19.45%)	1917 (19.61%)	
18,000 to 30,999	13,630 (27.88%)	2681 (27.42%)	2744 (28.06%)	2769 (28.32%)	2750 (28.12%)	2686 (27.47%)	
31,000 to 51,999	10,066 (20.59%)	1831 (18.72%)	2020 (20.66%)	2077 (21.24%)	2067 (21.14%)	2071 (21.18%)	
52,000 to 100,000	5464 (11.18%)	909 (9.30%)	1048 (10.72%)	1140 (11.66%)	1206 (12.33%)	1161 (11.87%)	
>100,000	1324 (2.71%)	179 (1.83%)	244 (2.50%)	288 (2.95%)	309 (3.16%)	304 (3.11%)	
Do not know/Prefer not to answer	8255 (16.88%)	1845 (18.87%)	1636 (16.73%)	1591 (16.27%)	1544 (15.79%)	1639 (16.76%)	
**Smoking status (%)**							<0.001
Never smoked	28,286 (57.85%)	5507 (56.31%)	5715 (58.45%)	5739 (58.69%)	5728 (58.58%)	5597 (57.24%)	
Ever smoked	20,461 (41.85%)	4244 (43.40%)	4023 (41.14%)	4014 (41.05%)	4031 (41.23%)	4149 (42.43%)	
Do not know/Prefer not to answer	145 (0.30%)	28 (0.29%)	40 (0.41%)	26 (0.27%)	19 (0.19%)	32 (0.33%)	
**Alcohol status (%)**							<0.001
Never drink	2204 (4.51%)	583 (5.96%)	482 (4.93%)	369 (3.77%)	393 (4.02%)	377 (3.86%)	
Previous drinker	1486 (3.04%)	404 (4.13%)	289 (2.96%)	262 (2.68%)	253 (2.59%)	278 (2.84%)	
Current drinker	45,157 (92.36%)	8779 (89.77%)	8998 (92.02%)	9139 (93.46%)	9124 (93.31%)	9117 (93.24%)	
Do not know/Prefer not to answer	45 (0.09%)	13 (0.13%)	9 (0.09%)	9 (0.09%)	8 (0.08%)	6 (0.06%)	
**BMI (kg/m^2^) (mean (SD))**	26.64 (4.64)	27.30 (5.03)	26.79 (4.70)	26.49 (4.48)	26.34 (4.40)	26.31 (4.51)	<0.001
**Energy intake (kcal/day) (mean [SD])**	1952 (460)	2019 (519)	1905 (441)	1903 (435)	1926 (430)	2009 (455)	<0.001
**Physical activity (MET-min/week) (mean [SD])**	2555 (2106)	2477 (2094)	2536 (2134)	2507 (2021)	2601 (2135)	2655 (2141)	<0.001
**Glycated haemoglobin (HbA1c) (mmol/mol) (mean [SD])**	36.44 (5.30)	36.81 (5.35)	36.53 (5.76)	36.46 (5.45)	36.21 (4.86)	36.21 (5.02)	<0.001
**Blood cholesterol (mmol/L) (mean [SD])**	6.10 (1.09)	6.11 (1.09)	6.12 (1.11)	6.12 (1.08)	6.11 (1.09)	6.06 (1.06)	<0.001
**Apolipoprotein E4 (APOE4) (%)**	11,272 (23.59%)	2115 (22.13%)	2243 (23.46%)	2259 (23.61%)	2346 (24.56%)	2309 (24.17%)	0.001
**Cardiovascular disease (%)**	1144 (2.34%)	249 (2.55%)	231 (2.36%)	232 (2.37%)	209 (2.14%)	223 (2.28%)	0.43
**Hypertension (%)**	13,754 (28.13%)	2969 (30.36%)	2756 (28.19%)	2720 (27.81%)	2645 (27.05%)	2664 (27.24%)	<0.001
**Depression (%)**	2452 (5.02%)	571 (5.84%)	487 (4.98%)	455 (4.65%)	461 (4.71%)	478 (4.89%)	<0.001
**Diabetes mellitus (%)**	1347 (2.76%)	281 (2.87%)	273 (2.79%)	265 (2.71%)	256 (2.62%)	272 (2.78%)	0.85
**Sleep duration = sleep duration ≥7 and ≤9 h (%)**	37,044 (75.77%)	7257 (74.21%)	7419 (75.87%)	7441 (76.09%)	7498 (76.68%)	7429 (75.98%)	0.001
**Red meat intake (serve/day)** **(mean [SD])**	0.28 (0.39)	0.20 (0.34)	0.25 (0.35)	0.29 (0.37)	0.32 (0.39)	0.36 (0.46)	<0.001
**Processed meat intake (serve/day) (mean [SD])**	0.42 (0.69)	0.49 (0.78)	0.45 (0.71)	0.42 (0.66)	0.39 (0.64)	0.35 (0.66)	<0.001
**Fish intake (serve/day) (mean [SD])**	3.58 (2.45)	2.58 (1.92)	3.11 (1.98)	3.47 (2.13)	3.91 (2.32)	4.80 (3.11)	<0.001
**Vegetable intake (serve/day)** **(mean [SD])**	2.63 (4.18)	2.20 (4.06)	2.48 (4.07)	2.50 (3.71)	2.79 (4.37)	3.16 (4.58)	<0.001
**Fruit intake (serve/day)** **(mean [SD])**	0.35 (0.46)	0.33 (0.47)	0.34 (0.45)	0.35 (0.45)	0.36 (0.45)	0.36 (0.49)	<0.001
**Sugar intake (g/day) (mean [SD])**	118.55 (43.10)	122.17 (48.37)	114.72 (40.35)	114.56 (39.98)	117.11 (40.31)	124.20 (44.96)	<0.001
**Vitamin C (mg/day) (mean [SD])**	160.68 (99.41)	126.53 (90.16)	145.52 (86.79)	158.19 (89.65)	171.41 (95.28)	201.74 (115.91)	<0.001
**Vitamin E (mg/day) (mean [SD])**	9.25 (3.84)	8.92 (4.18)	8.79 (3.51)	8.88 (3.50)	9.35 (3.62)	10.32 (4.10)	<0.001
**Salt intake (%)**							<0.001
Never/rarely	29,485 (60.31%)	5433 (55.56%)	5833 (59.65%)	6026 (61.62%)	6035 (61.72%)	6158 (62.98%)	
Sometimes	12,998 (26.59%)	2771 (28.34%)	2631 (26.91%)	2516 (25.73%)	2576 (26.34%)	2504 (25.61%)	
Usually	4926 (10.08%)	1161 (11.87%)	1024 (10.47%)	965 (9.87%)	918 (9.39%)	858 (8.77%)	
Always	1467 (3.00%)	413 (4.22%)	285 (2.91%)	268 (2.74%)	247 (2.53%)	254 (2.60%)	
Missing data	16 (0.03%)	1 (0.01%)	5 (0.05%)	4 (0.04%)	2 (0.02%)	4 (0.04%)	
**Fat intake (g/day) (mean [SD])**	72.13 (24.40)	80.98 (26.55)	72.83 (23.33)	69.90 (22.75)	68.69 (22.84)	68.25 (24.02)	<0.001
**Saturated fat intake (g/day) (mean [SD])**	27.68 (10.55)	32.96 (11.81)	28.27 (9.87)	26.64 (9.61)	25.69 (9.52)	24.85 (9.75)	<0.001
**Polyunsaturated fat intake (g/day) (mean [SD])**	13.38 (6.29)	14.39 (6.98)	13.42 (6.18)	12.97 (5.98)	12.90 (5.85)	13.20 (6.29)	<0.001
**Iron supplement use (%)**							0.041
No	48,682 (99.57%)	9726 (99.46%)	9740 (99.61%)	9742 (99.62%)	9736 (99.57%)	9738 (99.59%)	
Yes	210 (0.43%)	53 (0.54%)	38 (0.39%)	37 (0.38%)	42 (0.43%)	40 (0.41%)	

**Table 2 nutrients-15-00260-t002:** Risk for all-cause dementia associated with iron intake in females and males.

Female (n = 48,892, number of cases = 492)					
	**Energy-adjusted iron intake quintiles**				
	Q1	Q2	Q3	Q4	Q5
Person-years (py)	114,059.31	114,990.72	115,057.01	115,013.13	115,183.31
Case	110	107	88	96	91
Incidence rate (per 10^4^ py) (95%CI)	9.64 (8, 11.62)	9.3 (7.69, 11.24)	7.64 (6.2, 9.42)	8.35 (6.83, 10.20)	7.90 (6.43, 9.70)
M1: HR (95% CI)	1.3 (0.98, 1.72)	1.21 (0.91, 1.6)	Reference	1.08 (0.81, 1.44)	1.04 (0.77, 1.39)
*p*-value	0.068	0.192	-	0.608	0.809
M2: HR (95%CI)	1.22 (0.92, 1.63)	1.17 (0.88, 1.56)	-	1.04 (0.78, 1.4)	1.04 (0.77, 1.4)
*p*-values	0.175	0.27	-	0.789	0.8
M3: HR (95% CI)	1.14 (0.84, 1.54)	1.15 (0.87, 1.54)	-	0 (0, 1.42)	1.09 (0.8, 1.48)
M3: *p*-value	0.398	0.327	-	0.717	0.597
**Male (n = 41,213, number of cases = 560)**					
	**Energy-adjusted iron intake quintiles**				
	Q1	Q2	Q3	Q4	Q5
Person-years (py)	95,017.41	95,923.62	96,008.321	96,295.606	95,765.02
Case	115	113	107	90	135
Incidence rate (per 10^4^ py) (95%CI)	12.1 (10.08, 14.53)	11.78 (9.79, 14.16)	11.14 (9.22, 13.46)	9.35 (7.60, 11.49)	14.10 (11.91, 16.69)
M1: HR (95% CI)	1.38 (1.05, 1.82)	1.29 (0.98, 1.7)	1.21 (0.91, 1.6)	Reference	1.52 (1.16, 1.98)
*p*-value	0.022	0.074	0.192	-	0.002
M2: HR (95%CI)	1.29 (0.97, 1.71)	1.29 (0.97, 1.7)	1.19 (0.9, 1.59)	-	1.51 (1.15, 1.98)
*p*-values	0.075	0.077	0.225	-	0.003
M3: HR (95% CI)	1.37 (1.01, 1.86)	1.34 (1.01, 1.79)	1.21 (0.91, 1.62)	-	1.46 (1.11, 1.92)
M3: *p*-value	0.042	0.044	0.188	-	0.007

Abbreviations: Q, quintile; HR, hazard ratio; CI, confidence interval; py, person-years. Model 1 (M1): The hazard ratio was adjusted for age. Model 2 (M2): Additional adjusted for body mass index (BMI), smoking, alcohol intake, income, education, ethnicity, physical activity, blood cholesterol, sleep duration, energy intake, hypertension, cardiovascular disease, diabetes, depression and Apolipoprotein Ɛ4 (APOE4). Model 3 (M3): Additional adjusted for intake of salt, red meat, processed meat, fish, vegetable, fruit, sugar, vitamin C, vitamin E, fat, saturated fat, polyunsaturated fat, iron supplement.

## Data Availability

Restrictions apply to the availability of these data. Data was obtained from UK Biobank and are available at https://www.ukbiobank.ac.uk (accessed on 17 November 2022) with the permission of UK Biobank.

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
