# Peer review of "The Association between Dietary Iron Intake and Incidence of Dementia in Adults Aged 60 Years or over in the UK Biobank"

_nutrients, 2023, doi:10.3390/nu15020260_

Round 1
Reviewer 1 Report
The subject of this manuscript is very interesting as dementia is common in aged adults and knowledge of the associated factors may be useful to prevent it. Data were obtained from a Biobank population based cohort of approximately half a million people, of which after applying inclusion and exclusion criteria, more than 90.000 men and women aged 60 years or over were selected and included for this association study. Incidence of dementia (all-cause dementia, Alzheimer’s disease, vascular dementia) was recorded during 11.8 years.
Results show a U-shape association between iron intake and the incidence of dementia in both genders. Energy-adjusted iron intake quintiles were formed separately in men and women, and Q3 and Q4 in women in men respectively, presented the lowest incidence and were considered the reference quintiles. Different models were tested and finally it was concluded that excess intake of iron was associated with all-cause dementia and Alzheimer’s disease in men. In contrast, deficient iron intake was associated with all-cause dementia in women with hypertension.
I have several comments:
Abstract:
-Line 34. Although the meaning is clear, I suggest deleting “but not linear”. Therefore, the sentence is easier to understand: “A non-linear relationship between iron intake and….”.
-Line 38. Please indicate in the results summary that no association was found with excessive iron or deficient iron in women, except for the association between low iron and all-cause dementia in women with hypertension. This should be revised since in the present abstract the conclusion includes a sentence for women that was not mentioned before.
Introduction:
Line 65-70. Men have a higher iron status than women and the differences remain several years after menopause. Two articles mentioned here (Refs. 11 and 12) refer to iron excess that induces oxidative stress and tissue damage. The sentence should be rewritten.
The main mutations in the HFE gene are C282Y and H63D, these can induce iron overload or not depending on penetrance and many other factors. For instance, these mutations may protect menstruating women from iron deficiency anemia. There are many other mutations, some of them counteract the effects of the HFE mutations. References:
Am J Hum Genet. 2009 Jan;84(1):60-5. doi: 10.1016/j.ajhg.2008.11.011
Int J Mol Sci. 2014 Mar 6;15(3):4077-87. doi: 10.3390/ijms15034077
Line 66. Here it would be interesting to give the recommended dietary iron values for adult men and women.
Method:
-Ref 19 refers to the protocol of the UK-biobank. Please add complete identification of this source in the references section (Web access o similar). I see that this biobank collected information of people 40 to 69 years old. This means that there is a very narrow age range for the selection of subjects for the present study. This is important because incidence of dementia would be higher in older that younger adults. In fact, mean and SD age given on Table 1 confirms this limitation of the study.
-Another important factor is the use of supplements. Is this available?. When results indicate a relationship with excess of iron it is interesting to know if it was provided from the usual diet or the diet plus supplemental iron. Please, see below my suggestion on the concept of “excess” and “deficiency”.
Results:
Education level. Please explain the meaning of the levels, A, O, CSE, NVQ, etc. Otherwise, simplify the list.
The incidence of dementia was lower in women than men, about 1% compared with 1.36 cases, respectively. I think that this difference should be indicated.
Discussion:
The concept of iron deficiency and iron excess need some explanation. I miss a comment on which iron intake level appears to be more beneficial considering the recommended levels in men and women. I think that Q1 intake values are at the recommended levels for men and women older than 60 y while it seems that iron intakes higher than 13.7 mg/day and (Q3) and higher than 17.6 mg/day (Q4) are associated with “iron excess”. Please revise the text and consider this idea into account for the conclusion and the abstract.
Limitations of the study:
-The age of the participants, they were relatively young
-Iron intake is the main variable but no measure of iron status is given
Author Response
We are most grateful for the reviewers’ valuable and helpful comments, and for the opportunity to have our manuscript reconsidered with additional revisions. We have highlighted the revisions within the manuscript, in the hope that addresses the concerns and issues raised by the editors and reviewers
Line 33. Although the meaning is clear, I suggest deleting “but not linear”. Therefore, the sentence is easier to understand: “A non-linear relationship between iron intake and….”.
Response: Thank you for your advice. We have already deleted “but not linear”.
Line: 34 - 35
“A non-linear relationship between iron intake and incident dementia was observed in both males and females.”
-Line 41. Please indicate in the results summary that no association was found with excessive iron or deficient iron in women, except for the association between low iron and all-cause dementia in women with hypertension. This should be revised since in the present abstract the conclusion includes a sentence for women that was not mentioned before.
Response: Thanks to your suggestion, we have made some changes to our analysis model, therefore the outcome is changed.
Line: 42 - 45
“Conclusion Excessive iron intake (≥17.57 mg/day) is associated with a higher incidence of all-cause dementia in males and smoking status modified this association amongst males. Deficient iron intake (<10.93 mg/day) was associated with a higher incidence of all-cause dementia in females without a history of hypertension.”
Line 65-70. Men have a higher iron status than women and the differences remain several years after menopause. Two articles mentioned here (Refs. 11 and 12) refer to iron excess that induces oxidative stress and tissue damage. The sentence should be rewritten.
The main mutations in the HFE gene are C282Y and H63D, these can induce iron overload or not depending on penetrance and many other factors. For instance, these mutations may protect menstruating women from iron deficiency anemia. There are many other mutations, some of them counteract the effects of the HFE mutations. References:
Am J Hum Genet. 2009 Jan;84(1):60-5. doi: 10.1016/j.ajhg.2008.11.011
Int J Mol Sci. 2014 Mar 6;15(3):4077-87. doi: 10.3390/ijms15034077
Response: Thank you for your advice and valuable comments, we have added these references to the introduction section.
Line 72-81:
“These differences are attributed to gender-specific responses to iron metabolism gene variants, transferrin C2 (TfC2) and hemochromatosis (HFE) H63D and HFE C282Y, which are responsible for iron absorption, iron transportation and iron affinity regulation, respectively [1-3]. Previous studies demonstrated mutations on these are protective against iron deficiency anemia for reproductive-age females but not in males [3], and HFE C282Y mutations accompanied by haem iron intake is associated with higher ferritin levels in post-menopausal females [4]. Gender-specific responses to iron metabolism gene variants may be a theoretic underpinning for the association between iron intake and dementia in females and males separately.”
Line 66. Here it would be interesting to give the recommended dietary iron values for adult men and women.
Response: Thank you for your advice. We have given the recommended dietary iron values for adult males and females: 19.3-20.5 mg/day for adult males and 17.0-18.9 mg/day for adult females.
Line 70-72:
“The recommended dietary iron level is substantially different for males and females, with the average recommended dietary iron intake being 19.3-20.5 mg/day for adult males and 17.0-18.9 mg/day for adult females [5].”
Reference:
Beshaw T, Demssie K, Tefera M, Guadie A. Determination of proximate composition, selected essential and heavy metals in sesame seeds (Sesamum indicum L.) from the Ethiopian markets and assessment of the associated health risks. Toxicology Reports. 2022;9:1806-12.
Ref 19 refers to the protocol of the UK-biobank. Please add complete identification of this source in the references section (Web access o similar). I see that this biobank collected information of people 40 to 69 years old. This means that there is a very narrow age range for the selection of subjects for the present study. This is important because incidence of dementia would be higher in older that younger adults. In fact, mean and SD age given on Table 1 confirms this limitation of the study.
Response:
Thank you for your advice. We have added this limitation in the part of the discussion. However, most dementia incident cases occurred in the population aged 60 and above. In our analysis, among individuals with complete dietary data, only 183 dementia cases over 106,079 participants aged 60 and less have dietary data. In this study, 1,052 dementia incidents over 90,105 participants have been included. Likewise, previous studies conducted dementia research among adults aged 60 years old and older by using the UK-Biobank study. We hope this could eliminate your concern [6-8].
Reference:
Tai, X. Y., Veldsman, M., Lyall, D. M., Littlejohns, T. J., Langa, K. M., Husain, M., ... & Llewellyn, D. J. (2022). Cardiometabolic multimorbidity, genetic risk, and dementia: a prospective cohort study. The Lancet Healthy Longevity, 3(6), e428-e436.
Lourida, I., Hannon, E., Littlejohns, T. J., Langa, K. M., Hyppönen, E., Kuźma, E., & Llewellyn, D. J. (2019). Association of lifestyle and genetic risk with incidence of dementia. Jama, 322(5), 430-437.
Parra, K. L., Alexander, G. E., Raichlen, D. A., Klimentidis, Y. C., & Furlong, M. A. (2022). Exposure to air pollution and risk of incident dementia in the UK Biobank. Environmental Research, 209, 112895.
Line 425-426:
“Secondly, because of the small age range for subject selection in our investigation, this study has limited generalizability for those under the age of 60.”
Another important factor is the use of supplements. Is this available? When results indicate a relationship with excess of iron it is interesting to know if it was provided from the usual diet or the diet plus supplemental iron. Please, see below my suggestion on the concept of “excess” and “deficiency”.
Response:
Thank you for your consideration. We have further adjusted the iron supplementation covariates in model 3 and interpreted the iron intake data.
Line: 145-146:
“Iron intake for model analysis were extracted from dietary foods. Iron supplements were not included as part of iron intake.”
Education level. Please explain the meaning of the levels, A, O, CSE, NVQ, etc. Otherwise, simplify the list.
Response:
Thank you for your advice, we have added more descriptions for the education level. This education level comes from the UK Biobank degree level (Data-Coding 100305) https://biobank.ndph.ox.ac.uk/showcase/field.cgi?id=6138.
Line: 156-160:
“Education (high level [college or university degree (Level 6-8 education qualifications)], intermediate level [A levels/AS levels or equivalent, O levels/GCSEs or equivalent, CSEs or equivalent, NVQ or HND or HNC or equivalent, and other professional qualifications (Level 2-5 education qualifications)], and low level [none of the aforementioned])”
The incidence of dementia was lower in women than men, about 1% compared with 1.36 cases, respectively. I think that this difference should be indicated.
Response:
Thank you for your advice. We have already pointed it out.
Line: 244-246:
“The incidence of all-cause dementia was higher in males (Incidence rate [IR]: 11.7 per 104 person-year [py]) than in females (IR: 8.6 per 104 py).”
The concept of iron deficiency and iron excess need some explanation. I miss a comment on which iron intake level appears to be more beneficial considering the recommended levels in men and women. I think that Q1 intake values are at the recommended levels for men and women older than 60 y while it seems that iron intakes higher than 13.7 mg/day and (Q3) and higher than 17.6 mg/day (Q4) are associated with “iron excess”. Please revise the text and consider this idea into account for the conclusion and the abstract.
Response:
Thank you for your advice. The recommended energy-adjusted iron levels are (Q3) 12.4-13.71 mg/day for women and (Q4) 15.73-17.57 mg/day for men, and these levels are consistent with previous studies where "recommended dietary iron levels varied considerably between men and women, with mean recommended dietary iron intakes of 19.3-20.5 mg/day for men and 17.0-18.9 mg/day for women [5]” that higher level of iron intake is recommended in males than in females. the recommended iron intake for men is higher than the level for women. The reason for using Q3 in females and Q4 in males is based on the lowest incidence rate in the quintile of energy-adjusted iron intake. A more intuitive hazard ratio can be observed when comparing the incidence with the other quintiles.
Line: 35-37:
“The lowest incidence rates were observed in the higher iron intake quintile (Q4: ≥15.73, <17.57 mg/day) for males, and the intermediate iron intake quintile (Q3: ≥12.4, <13.71 mg/day) for females.”
Line: 70-72:
“The recommended dietary iron level is substantially different for males and females, with the average recommended dietary iron intake being 19.3-20.5 mg/day for adult males and 17.0-18.9 mg/day for adult females [5].”
Line: 437-440:
“Excessive iron intake (≥17.57 mg/day) is associated with a higher incidence of all-cause dementia in males and history of smoking further increased this risk. Deficient iron intake (<10.93 mg/day) was associated with a higher incidence of all-cause dementia in females without a history of hypertension.”
- Bartzokis, G., et al., Gender and iron genes may modify associations between brain iron and memory in healthy aging. Neuropsychopharmacology, 2011. 36(7): p. 1375-1384.
- Zimmermann, M.B. and R.F. Hurrell, Nutritional iron deficiency. Lancet, 2007. 370(9586): p. 511-520.
- Blanco-Rojo, R., et al., Influence of diet, menstruation and genetic factors on iron status: a cross-sectional study in Spanish women of childbearing age. Int J Mol Sci, 2014. 15(3): p. 4077-87.
- Cade, J.E., et al., Diet and genetic factors associated with iron status in middle-aged women. Am J Clin Nutr, 2005. 82(4): p. 813-20.
- Beshaw, T., et al., Determination of proximate composition, selected essential and heavy metals in sesame seeds (Sesamum indicum L.) from the Ethiopian markets and assessment of the associated health risks. Toxicology Reports, 2022. 9: p. 1806-1812.
- Tai, X.Y., et al., Cardiometabolic multimorbidity, genetic risk, and dementia: a prospective cohort study. The Lancet Healthy Longevity, 2022. 3(6): p. e428-e436.
- Lourida, I., et al., Association of lifestyle and genetic risk with incidence of dementia. Jama, 2019. 322(5): p. 430-437.
- Parra, K.L., et al., Exposure to air pollution and risk of incident dementia in the UK Biobank. Environmental Research, 2022. 209: p. 112895.

Reviewer 2 Report
In this manuscript, authors investigated the association between dietary iron intake and cognitive impairment. They investigated the association between dietary iron intake and dementia in males and females. Their analysis shows excessive iron intake is associated with a higher incidence of all-cause dementia and AD in males. Smoking status modified the association in males. Deficient iron intake was associated with a higher incidence of all-cause dementia in females with a history of hypertension.
I find this report impressive, data analysis are appropriately designed and results were conclusive.
I have some concerns noted below:
11) While analyzing such a large cohort authors must point all limitations of the study/analysis.
22) There are some many English typos/ errors (e.g. “study is did not”, line 371) must be addressed. Proof reading is required.
This work would be a good addition to the field.
I would recommend this article for publication after revision.
Author Response
We are most grateful for the reviewers’ valuable and helpful comments, and for the opportunity to have our manuscript reconsidered with additional revisions. We have highlighted the revisions within the manuscript, in the hope that addresses the concerns and issues raised by the editors and reviewers
- While analyzing such a large cohort authors must point all limitations of the study/analysis.
Response:
Thank you for your advice. We have pointed out the limitations of our study more comprehensively.
Line: 431-445:
“Some limitations should also be acknowledged. First, more than 97.5% of data were obtained from a White Caucasian background leading to poor generalizability to other ethnicities. Geographic differences may also exist between the United Kingdom and other nations, considering the varied prevalence of illness complications and levels of primary health care programmes. Secondly, because of the small age range for subject selection in our investigation, this study has limited generalizability for those under the age of 60. Thirdly, this study used a self-reported questionnaire to calculate iron intake, and this likely leads to measurement errors in iron consumption. Fourth, people who developed dementia may have delayed reporting or diagnosis, and misclassification bias may have reduced reporting. Likewise, the misclassification of dementia subtypes needs to be considered as AD and VD often has mixed pathology and pathophysiology [1]. Potentially, medications may interfere with iron absorption and impact dementia pathogenesis. Lastly, there may be insufficient power to establish the synergistic effects or modification effects of the aforementioned medical and physiological conditions. Further studies are needed before a definitive conclusion can be drawn.”
- There are some many English typos/ errors (e.g. “study is did not”, line 371) must be addressed. Proof reading is required.
Response:
Thank you for your advice. We have modified it.
Line: 345-347:
“These findings imply iron intake should reduce with age to minimize modifiable risk, although it should be noted those with stroke history were not excluded from the current analysis.”
- Kalaria, R.N., The pathology and pathophysiology of vascular dementia. Neuropharmacology, 2018. 134: p. 226-239.

Reviewer 3 Report
Here, these workers have mined the UK Biobank database to interrogate the relationships between dietary iron intake and the incidence of dementia in both males and females. 41,213 males and 48,892 females aged 60 years or over were included from the 500,000 plus available as they had an adequate and annual dietary record self recorded over the first four years. These workers also examined whether an association was modified by age and a number of medical conditions that had subsequently developed. Incident all cause, Alzheimer, and vascular dementia was ascertained from hospital inpatient records and the death register until April 2021. The association between iron intake and incident dementia was analysed using Cox proportional regression models and hazard ratio curves were constructed. 560 males and 492 females developed dementia over a mean 12 year follow-up. A U shaped relationship between iron intake and incident dementia was observed in both males and females. The hazard ratio for all-cause dementia in males associated with excessive iron intake was 1.49 but was mainly due to previous smokers. There was no increased risk found in females. Deficient dietary iron intake was associated with an increased risk of all-cause dementia in females if they were also hypertensive.
This study extends smaller surveys that have studied the link between dietary iron intake and Parkinson’s disease, dementia, and cognitive function in humans. Results have been inconsistent across these studies. This much larger study has reported an increased HR of dementia from excessive iron intake in males but the risk is still small compared to the risks associated with past stroke, hypertension, and diabetes. Unexpectedly, past but not current smoking modified this risk which is counter intuitive. A difficulty with mining the UK Biobank database is that dietary records are based on subject recall and the diagnosis of dementia is dependent on medical records. While this allows the incidence of all cause dementia to be assessed, one cannot reliably separate Alzheimer and vascular dementia from records alone as these dementias usually have mixed pathology. While the calculated excessive iron intake increased the risk of dementia in males, this was presumably associated with high red and processed meat intake and so could reflect a high fat and salt diet as much as iron intake. In females this increased HR was not seen and their raised risk from iron deficiency when hypertensive is most likely to have reflected the latter complication. Finally, the UK Biobank population is essentially white and educated and so findings will not necessarily be generalisable to other ethnic groups such as Asians where diabetes, hypertension, and cerebrovascular disease are more prevalent. Having said that, the findings are certainly of interest and cutting back on red and processed meat is to be recommended.
Author Response
We are most grateful for the reviewers’ valuable and helpful comments, and for the opportunity to have our manuscript reconsidered with additional revisions. We have highlighted the revisions within the manuscript, in the hope that addresses the concerns and issues raised by the editors and reviewers
Response:
Thank you for your comment. No significant association could be observed in male smokers, probably due to the small sample size (n: 2988, cases: 37). Although the prevalence of iron overdose quintile (men: Q4: 16.63/104 py) among current smokers is severely higher than the reference quintile (men: Q3: 10.56/104 py). This is not sufficient to address the true association. Therefore, with this situation in mind, we combined current and former smokers into one group for this subgroup analysis. Unsurprisingly, we observed a clear association in all three models.
While this allows the incidence of all cause dementia to be assessed, one cannot reliably separate Alzheimer and vascular dementia from records alone as these dementias usually have mixed pathology.
Response:
Thank you for your comments and consideration, your concern is critical. We have added this issue to our restrictions section.
Line: 430-431:
“Likewise, the misclassification of dementia subtypes needs to be considered as AD and VD often has mixed pathology and pathophysiology (6).”
While the calculated excessive iron intake increased the risk of dementia in males, this was presumably associated with high red and processed meat intake and so could reflect a high fat and salt diet as much as iron intake.
Response:
Thanks for your comment and your consideration, your concern is critical and inspiring. We added a model (Model 3) to our analysis. We hope this can eliminate your concerns.
We generate a new Model 3 based on model 2, which is further adjusted for the intake of red meat, processed meat, fish, vegetable, fruit, sugar, vitamin C, vitamin E, fat, saturated fat, polyunsaturated fat, and iron supplement. The results remained significant.
In females this increased HR was not seen and their raised risk from iron deficiency when hypertensive is most likely to have reflected the latter complication.
Response:
The result is inconsistent with Shi et al., who have insufficient evidence to demonstrate a modification effect by hypertension on the association between dietary iron and dementia and cognitive impairment (23). This may be because the analysis was not conducted separately for males and females in the first place. However, there is evidence that showed that an increased level of blood pressure is associated with an increased risk of dementia (63), and the prevalence of iron deficiency is higher in people with idiopathic pulmonary hypertension compared with disease-free patients (64). Therefore, a plausible hypothesis is that insufficient iron intake is associated with an increased risk of hypertension, and consequently enhanced risk of developing dementia.
Finally, the UK Biobank population is essentially white and educated and so findings will not necessarily be generalisable to other ethnic groups such as Asians where diabetes, hypertension, and cerebrovascular disease are more prevalent.
Response:
Thanks for your comment and your consideration, your comment is critical and we have updated your comment in our limitation section.
Line: 421-435:
“Some limitations should also be acknowledged. First, more than 97.5% of data were obtained from a White Caucasian background leading to poor generalizability to other ethnicities. Geographic differences may also exist between the United Kingdom and other nations, considering the varied prevalence of illness complications and levels of primary health care programmes. Secondly, because of the small age range for subject selection in our investigation, this study has limited generalizability for those under the age of 60. Thirdly, this study used a self-reported questionnaire to calculate iron intake, and this likely leads to measurement errors in iron consumption. Fourth, people who developed dementia may have delayed reporting or diagnosis, and misclassification bias may have reduced reporting. Likewise, the misclassification of dementia subtypes needs to be considered as AD and VD often has mixed pathology and pathophysiology (6). Potentially, medications may interfere with iron absorption and impact dementia pathogenesis. Lastly, there may be insufficient power to establish the synergistic effects or modification effects of the aforementioned medical and physiological conditions. Further studies are needed before a definitive conclusion can be drawn.“
Having said that, the findings are certainly of interest and cutting back on red and processed meat is to be recommended.
Response:
Thanks for your comment and your recommendation, we added it to the Model 3.
